# AMPKα Subunit Ssp2 and Glycogen Synthase Kinases Gsk3/Gsk31 are involved in regulation of sterol regulatory element-binding protein (SREBP) activity in fission yeast

Hao Miao[1], Qiannan Liu[1], Guanglie Jiang[1], Wen Zhang[1], Kun Liu[1], Xiang Gao[1], Yujie Huo[1], Si Chen[1], Toshiaki Kato[2], Norihiro Sakamoto[2], Takayoshi Kuno[1,2], Yue Fang[1] *

1 Department of Microbial and Biochemical Pharmacy, School of Pharmacy, China Medical University, Shenyang, Liaoning Province, China, 2 Division of Food and Drug Evaluation Science, Kobe University Graduate School of Medicine, Kobe, Japan

* yfang@cmu.edu.cn

**Data Availability Statement:** All relevant data are within the paper and its Supporting Information files.

## Abstract

Sterol regulatory element-binding protein (SREBP), a highly conserved family of membrane-bound transcription factors, is an essential regulator for cellular cholesterol and lipid homeostasis in mammalian cells. Sre1, the homolog of SREBP in the fission yeast *Schizosaccharomyces pombe* (*S. pombe*), regulates genes involved in the transcriptional responses to low sterol as well as low oxygen. Previous study reported that casein kinase 1 family member Hhp2 phosphorylated the Sre1 N-terminal transcriptional factor domain (Sre1N) and accelerated Sre1N degradation, and other kinases might exist for regulating the Sre1 function. To gain insight into the mechanisms underlying the Sre1 activity and to identify additional kinases involved in regulation of Sre1 function, we developed a luciferase reporter system to monitor the Sre1 activity through its binding site called SRE2 in living yeast cells. Here we showed that both ergosterol biosynthesis inhibitors and hypoxia-mimic CoCl₂ caused a dose-dependent increase in the Sre1 transcription activity, concurrently, these induced transcription activities were almost abolished in Δ*sre1* cells. Surprisingly, either AMPKα Subunit Ssp2 deletion or Glycogen Synthase Kinases Gsk3/Gsk31 double deletion significantly suppressed ergosterol biosynthesis inhibitors- or CoCl₂-induced Sre1 activity. Notably, the Δ*ssp2*Δ*gsk3*Δ*gsk31* mutant showed further decreased Sre1 activity when compared with their single or double deletion. Consistently, the Δ*ssp2*Δ*gsk3*Δ*gsk31* mutant showed more marked temperature sensitivity than any of their single or double deletion. Moreover, the fluorescence of GFP-Sre1N localized at the nucleus in wild-type cells, but significantly weaker nuclear fluorescence of GFP-Sre1N was observed in Δ*ssp2*, Δ*gsk3*Δ*gsk31*, Δ*ssp2*Δ*gsk3*, Δ*ssp2*Δ*gsk31* or Δ*ssp2*Δ*gsk3*Δ*gsk31* cells. On the other hand, the immunoblot showed a dramatic decrease in GST-Sre1N levels in the Δ*gsk3*Δ*gsk31* or the Δ*ssp2*Δ*gsk3*Δ*gsk31* cells but not in the Δ*ssp2* cells. Altogether, our findings suggest that Gsk3/Gsk31 may regulate Sre1N degradation, while Ssp2 may regulate not only the degradation of Sre1N but also its translocation to the nucleus.

**Funding:** This work was supported by grants from the Natural Science Foundations of Liaoning Province of China (No. 2019-MS-384) to YF. This work was also supported by Research Fund from PhD research startup foundation of Liaoning Province, China (No. 2019-BS-284) to SC. The funders had no role in study design, data collection and analysis, decision to publish, or preparation of the manuscript.

**Competing interests:** The authors have declared that no competing interests exist.

## Introduction

Sterol homeostasis is essential for eukaryotic cells to maintain the normal structure and fluidity of cell membrane as well as to regulate the function of membrane proteins and sterol synthesis. Sterol regulatory element binding protein (SREBP), a subfamily of basic helix-loop-helix leucine zipper (bHLH-LZ) transcription factors that are widely conserved from fungi to human, is an important factor that regulates sterol levels in cells [1, 2]. SREBP, as a non-activated precursor protein synthesized in the endoplasmic reticulum (ER), consists of an N-terminal transcription factor domain and a C-terminal domain that forms a complex with a sterol sensing protein, SREBP cleavage activating protein (Scap) [3]. Under sterol replete conditions, Scap binds cholesterol, and the SREBP-Scap complex is retained in the ER [4]. Upon sterol depletion, Scap undergoes a conformational change and SREBP-Scap enters COPII vesicles for transport to the Golgi [5, 6], and then SREBP is cleaved sequentially in the Golgi, resulting in the release of the SREBP transcription factor domain, which then translocate into the nucleus and bind to the specific DNA sequence (sterol regulatory element, SRE) of the target genes involved in sterol synthesis [7].

In fission yeast, Sre1, the homologue of mammalian SREBP, is not only a factor for controlling sterol homeostasis but also a principal regulator of low oxygen gene expression [2, 8]. It has been reported that, upon conditions of low oxygen, ergosterol biosynthesis decreases, and Scp1, the homologue of Scap, transports Sre1 from the ER to the Golgi where Sre1 is proteolytically cleaved, releasing the active Sre1 N-terminal transcription factor fragment (Sre1N). Under low oxygen or low sterols conditions, the Sre1-Scp1 transports and cleavages at the Golgi increase dramatically. After released, Sre1N enters the nucleus and promotes transcription of the target genes involved in sterol synthesis as well as oxygen responsive genes. Upon reintroduction of oxygen or sterol, Sre1N degradation is accelerated through a proteasome-dependent pathway, allowing rapid down-regulation of Sre1N [9, 10]. Factors controlling Sre1 cleavage and activation have been largely studied [11, 12], but the mechanism underlying Sre1 degradation remains not fully understand.

The initial characterization of the fission yeast SREBP pathway revealed that the activated transcription factor Sre1N could be hyper-phosphorylated, indicating potential regulation of sterol homeostasis by kinases [8]. Recent studies have shown that a highly conserved casein kinase 1 family member Hhp2 phosphorylates Sre1N and accelerates Sre1N degradation, but it seems like that Hhp2 is not the sole Sre1N kinase. Sre1N contains at least 22 phosphorylated serine and threonine residues [13], suggesting that additional kinases might be involved in Sre1 activity regulation.

Here, our studies focus on whether additional protein kinases are involved in regulation of Sre1 activity. We monitored the transcriptional activity of Sre1 in living cells by using a luciferase reporter system with three tandem repeats of SRE2 fused to firefly luciferase gene. We found that ergosterol biosynthesis inhibitors- or hypoxia-mimic $CoCl_2$-induced Sre1 transcriptional activity was significantly suppressed in AMPKα Subunit Ssp2 deletion or Glycogen Synthase Kinases Gsk3/Gsk31 double deletion as well as double deletion of Ssp2 and Gsk3 or Gsk31, respectively. In particular, the nuclear fluorescence of GFP-Sre1N were dramatically reduced in these deletion cells. In addition, the immunoblot showed a dramatic decrease in Sre1N levels in the Δ*gsk3*Δ*gsk31* or Δ*ssp2*Δ*gsk3*Δ*gsk31* cells but not in the Δ*ssp2*, Δ*ssp2*Δ*gsk3* or Δ*ssp2*Δ*gsk31* cells. Our findings reveal the involvement of AMPKα Subunit Ssp2 and Glycogen Synthase Kinases Gsk3/31 in regulation of SREBP activity in fission yeast, which may pave a way for further studying similar mechanisms in higher eukaryotes.

## Materials and methods

### Strains, media, and genetic and molecular biology methods

*S. pombe* strains used in this study are listed in Table 1. The normal minimal medium EMM (Edinburgh minimal medium), the complete medium yeast extract-peptone-dextrose (YPD) and the rich yeast extract with supplements (YES) have been described previously [14]. Standard genetic and recombinant-DNA methods [15] were used except where noted. Gene disruptions are denoted by lower-case letters representing the disrupted gene followed by two colons and the wild type gene marker used for disruption (for example, *sre1::ura4*[+]). Gene disruptions are abbreviated by the gene preceded by Δ (for example, Δ*sre1*). Proteins are denoted by Roman letters and only the first letter is capitalized (for example, Sre1) [16].

### Plasmids constructions

A multicopy plasmid (pKB7665) containing the *nmt1* promoter without its *cis* element, three tandem repeats of SRE2-like sequence (ATCACCCCAT) which is the binding core of the Sre1 transcriptional activator identified in the *sre1*[+] promoter, and the destabilized luciferase from pGL3 (R2.2) version containing PEST, CL1, and AU-rich repeats was constructed as described previously [17], except that the CDRE oligonucleotides were replaced by the SRE2-like oligonucleotides (sense, 5′-GGC TTA TCA CCC CAT ATA CAA TCA CCC CAT ATA CAC AAT CAC CCC ATA TGC AC-3′; antisense, 5′-TCG AGT GCA TAT CAC CCC ATT GTG TAT ATC ACC CCA TTG TAT ATC ACC CCA TAA GCC TGC A-3′, SRE2-like sequence underlined). Then, an integration 3×SRE2::luc (R2.2) plasmid was constructed by inserting the SRE2-like oligonucleotides and *arg1*[+] into pBC SK(+) (Stratagene) to give pSY291 [18]. The multicopy plasmid and integration plasmid of 3×SRE2::luc (R2.2) were all used for real-time monitoring assay of Sre1-mediated transcriptional activity.

The truncated fragment of *sre1*[+] gene, encoding the active N-terminal transcription factor domain of Sre1 (Sre1N) was amplified by PCR with the genomic DNA of *S. pombe* as a template. The sense primer used for PCR was 5′-CGC GGA TCC ATG CAA AGC TCA ATT

**Table 1. Strains used in this study.**

| Strain | Genotype | Reference |
|---|---|---|
| HM123 | *h⁻ leu1-32* | Our stock |
| KP1737 | *h⁻ leu1-32 ura4-D18 gsk3::ura4⁺ gsk31::KanMX₆* | [27] |
| KP1813 | *h⁻ leu1-32 ura4-D18 gsk3::ura4⁺* | [27] |
| KP2101 | *h⁻ leu1-32 arg1* | [18] |
| KP2310 | *h⁻ leu1-32 ura4-D18 gsk31::KanMX₆* | [27] |
| KP3089 | *h⁻ leu1-32 ura4-D18 ssp2::ura4⁺* | [27] |
| KP4275 | *h⁻ leu1-32 ura4-D18 ssp2::ura4⁺ gsk31::KanMX₆* | [27] |
| KP4283 | *h⁻ leu1-32 ura4-D18 ssp2::ura4⁺gsk3::ura4⁺* | [27] |
| KP4304 | *h⁻ leu1-32 ura4-D18 asn1::loxp ssp2::asn1⁺* | This study |
| KP5683 | *h⁺ his2 leu1-32 ura4-D18 asn1::loxp gsk3::ura4⁺ gsk31::KanMX₆* | This study |
| KP6544 | *h⁻ leu1-32 ura4-D18 sre1::ura4⁺* | This study |
| CM109 | *h⁻ leu1-32 scp1::KanMX₄* | This study |
| CM125 | *h⁻ leu1-32 hhp2::KanMX₄* | This study |
| CM134 | *h⁻ leu1-32 ura4-D18 asn1::loxp ssp2::asn1 gsk3::ura4⁺ gsk31::KanMX₆* | This study |
| CM150 | *h⁻ leu1-32 arg1::arg1-3×SRE2-Luciferase(type(2.2))* | This study |
| CM156 | *h⁻ leu1-32 ura4-D18 arg1 sre1::ura4⁺* | This study |
| CM172 | *h⁻ leu1-32 ura4-D18 sre1::ura4 arg1::arg1-3×SRE2-Luciferase(type(2.2))* | This study |

CCG -3' (*Bam*HI site are underlined), and the antisense primer was 5'-ACG CGT CGA CTT ATG GAG ACA TAA GAA AAG-3' (*Sal*I site are underlined). The amplified product was digested with *Bam*HI/*Sal*I, and the resulting fragment was subcloned into Bluescript SK (+) (Stratagene, USA). To assess the subcellular localization and the total protein levels of Sre1N, the truncated fragment of *sre1*+ gene which encoding N-terminal transcriptional factor domain was ligated to the C terminus of the GFP or GST expressing GFP-Sre1N or GST-Sre1N and subcloned into the pREP1 expression vector containing a thiamine-repressible *nmt1* promoter [19]. Expression was repressed by the addition of 4 μM thiamine to EMM.

## Real-time monitoring assay of Sre1-mediated transcriptional activity

The multicopy 3×SRE2::luc (R2.2) reporter plasmid pKB7665 was transformed into fission yeast wide-type cells and mutations to perform the luciferase reporter assays as described previously [20]. In addition, to obtain the chromosome-borne 3×SRE2::luc (R2.2), the integration reporter plasmid linearized with *Stu*I was integrated into the chromosome at the *arg1*+ locus of both KP2101 (*h⁻ leu1-32 arg1*) and CM156 (*h⁻ leu1-32 ura4-D18 arg1 sre1::ura4*+) as described previously [18]. The cells transformed with the multicopy reporter plasmid or the chromosome-borne cells were selected and cultured in EMM or EMM with leucine to midlog phase at 30˚C and recovered by centrifugation respectively. Then the cells were resuspended in refresh EMM or EMM with leucine containing different concentrations of drugs. Luciferin was used as a substrate for *Firefly* luciferase, and yielding luminescence was detected using a luminometer (AB-2350; ATTO Co., Tokyo, Japan) at 1-min intervals and reported as relative light units (RLU).

## Gene deletion

To delete the *sre1*+ gene, a one-step gene disruption by homologous recombination [21] was performed. The *sre1*::*ura4*+ disruption was constructed as follows. The cloned open reading frame of the *sre1*+ gene in pBluescript SK (Stratagene) was digested with *Hind*III, and the resulting fragment containing the *sre1*+ gene was subcloned into the *Hind*III site of the pBluescript vector. Then a *Bam*HI fragment containing the *ura4*+ gene was inserted into the *Bam*HI site of the previous construct, causing the interruption of the open reading frame. The fragment containing the disrupted *sre1*+ gene was transformed into haploid cells. Stable integrants were selected on medium lacking uracil, and disruption of the gene was checked by Southern blotting.

The deletion of either *scp1*+ or *hhp2*+ gene with a genetic background of *h⁺ leu1-32 ura4-D18 ade6-M210 or M216* and the *KanMX* cassette was purchased from BioNEER (South Korea) [22]. We constructed *scp1* or *hhp2* deletion cells that were not auxotrophic for uracil and adenine by the genetic cross between wild-type cells HM123 and the above strains to make CM109 or CM125, respectively (Table 1). We constructed Δ*ssp2*Δ*gsk3*Δ*gsk31* triple deletion by the genetic cross between KP4304 (*h⁻ leu1-32 ura4-D48 asn1::loxp ssp2::asn1*+) and KP5683 (*h⁺ his2 leu1-32 ura4-D48 asn1::loxp gsk3::ura4*+ *gsk31::KanMX₆*) to make CM134 (*h⁻ leu1-32 ura4-D48 asn1::loxp ssp2::asn1*+*gsk3::ura4*+ *gsk31::KanMX₆*) (Table 1).

## Fluorescence microscopy

Cells transformed with pREP1-GFP-Sre1N were grown in EMM medium with 4 μM thiamine to attenuate the expression for 16 h at 30˚C, GFP-Sre1N was detected by its own fluorescence expressed in living cells by fluorescence microscope using a Nikon Eclipse Ni-U microscope equipped with a DS-Qi2 camera (Nikon Instruments Inc., Japan). For measurement of fluorescence intensities, images of the cells expressing GFP-Sre1N were taken using an oil-immersion

objective lens (UApo 100×, NA 1.3, Nikon) at NA = 0.65; The best-focused image of the eight optical sections was selected for quantification. Quantification of fluorescence intensities was determined as follows. Specify a region including the nucleus, and calculate the mean fluorescence intensity ($Mean_N$) and area values ($Area_N$) of the nuclear region. Calculate the mean fluorescence intensity in a cytoplasmic region of the same cell as the background mean fluorescence intensity ($Mean_B$). Calculate GFP-Nucleus fluorescence intensity on NE ($F_{NE}$) according to the following formula: $F_N = (MeanN—Mean_B) \times Area_N$. Measure the fluorescence intensities of about 50 cells in each strain, and calculate average values and standard deviations. Wide-type cells fluorescence intensity was set to one in calculating relative fluorescence intensities [23, 24].

## Cell extract preparation and immunoblot analysis

For the analysis of the total Sre1N protein in various mutants, whole cell extracts were prepared from cultures of wild-type cells or mutants harboring pREP1-GST-Sre1N plasmid grown at 30˚C to mid-log phase. Total cell lysates were prepared as follows. Approximately $2 \times 10^7$ cells were resuspended in 500 μl of homogenizing buffer (92.5% 2N NaOH, 7.5% β-mercaptoethanol). After cooling on ice for 10 min, the proteins were precipitated by the addition of 500 μl of 50% trichloroacetic acid. Then, cellular debris was removed by centrifugation at 14000 rpm for 5 min. The resulting protein extracts were subjected to sodium dodecyl sulfate-polyacrylamide gel electrophoresis (SDS-PAGE) [25]. We used separating acrylamide gels (10.0%) with a mono/bis ratio of 29:1 to detect Sre1N. Purified polyclonal anti-GST was used as the primary antibodies, and goat anti-rabbit immunoglobulin G (IgG) Fc fragments was used as the secondary antibodies. Enhanced chemiluminescence was used for immunodetection on the membrane.

## Statistical analyses

Quantitative data were expressed as means ± S.D. Multiple comparisons were statistically analyzed by one-way ANOVA followed by Tukey's test. The difference was considered to be significant, if P value is less than 0.05. All statistical analyses were performed using the SPSS 16.0 software package (SPSS, Inc., Chicago, IL, USA).

## Results

### Real-time monitoring of Sre1 activity in living cells

Mammalian SREBPs bind a 10-bp DNA sequence in the promoters of target genes, called the sterol regulatory element (SRE) [7]. In fission yeast, two DNA elements (SRE2 and SRE3) in the promoter of sre1+ gene were identified to be necessary and sufficient for oxygen-dependent, Sre1-dependent transcription, but the binding of Sre1 to SRE2 was stronger than to SRE3 [26]. In order to monitor the real-time Sre1 activity for further studying the regulation mechanisms underlying the Sre1 activity, we constructed multicopy reporter plasmid containing three tandem repeats of SRE2 fused to *firefly* luciferase gene. As shown in Fig 1A–1C, ergosterol biosynthesis inhibitors including clotrimazole (CLZ), terbinafine (TER), and fenpropimorph (FEN), increased the luciferase activity with a peak rise at about 12–13 hours through the SRE2 motif in a dose-dependent manner (Fig 1A–1C). Since it is known that Sre1 can bind to the SRE2 motif for its transcriptional control, we examined the involvement of Sre1 in ergosterol biosynthesis inhibitors-induced activation of SRE2 reporter. The results showed that these ergosterol biosynthesis inhibitors-induced increase in SRE2 reporter activity was completely abolished in Δsre1 cells (Fig 1D–1F), suggesting that the multicopy reporter

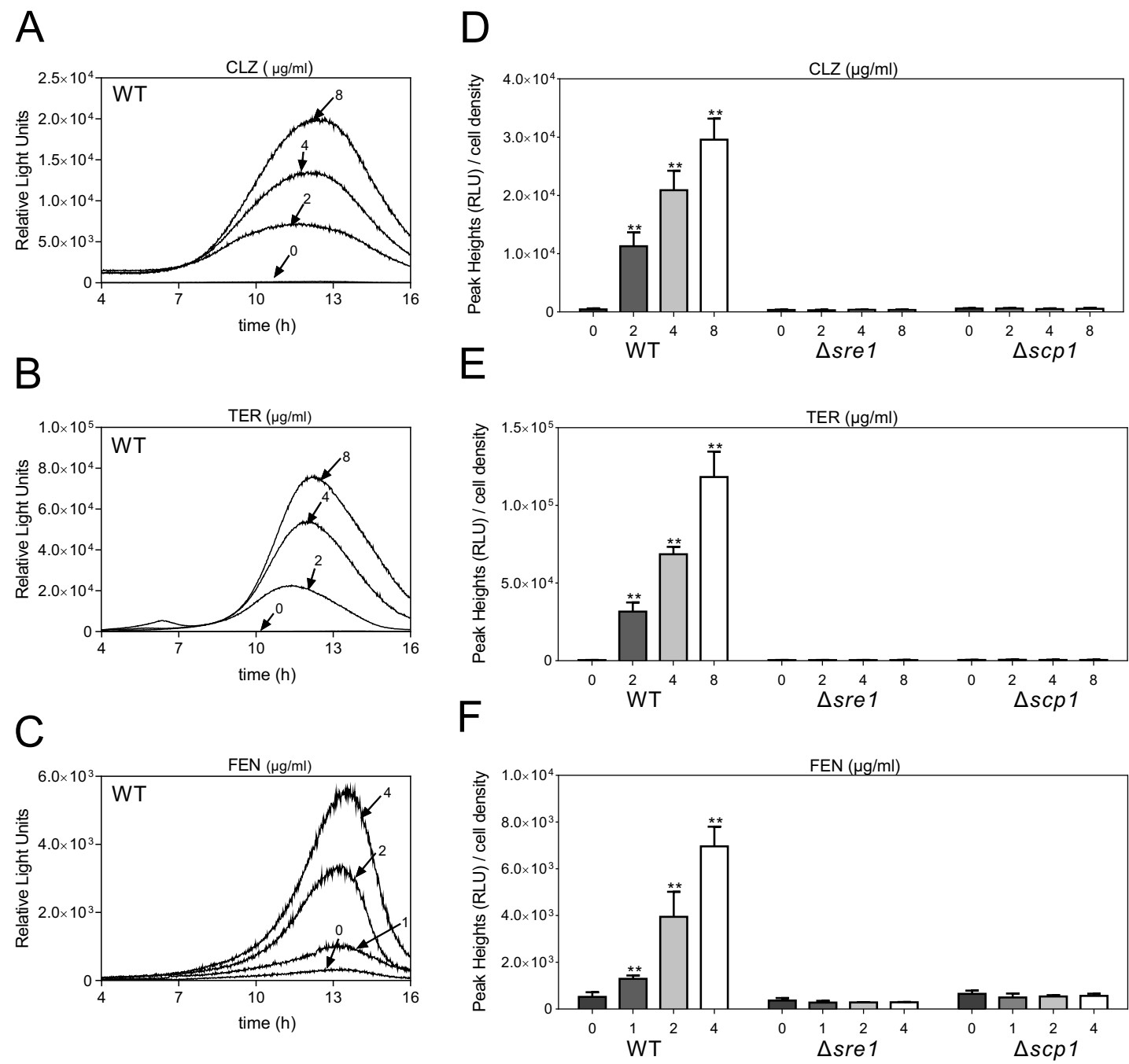

**Fig 1. Real-time monitoring of Sre1 activity in living cells.** (A) CLZ (2 μg/ml to 8 μg/ml), (B) TER (2 μg/ml to 8 μg/ml) and (C) FEN (1 μg/ml to 4 μg/ml) induced a marked increase in Sre1 transcriptional activity. Wild-type cells harboring the multicopy 3×SRE2::luc (R2.2) reporter plasmid were cultured and assayed as described in materials and methods. Y-axis values are the relative light units (RLU) of each sample. The data shown are representative of multiple experiments. (D-F) CLZ (2 μg/ml to 8 μg/ml), TER (2 μg/ml to 8 μg/ml) and FEN (1 μg/ml to 4 μg/ml)-induced transcriptional activity was almost completely abolished in Δ*sre1* and Δ*scp1* cells. The Δ*sre1* and Δ*scp1* cells harboring the multicopy reporter plasmid were cultured and assayed as described in Fig 1A-1C. Y-axis values are the relative light units (RLU) of peak height normalized to cell density (OD660) of each sample at the peak time. The data were averaged from three independent experiments, and each sample was done in triplicate. Error bars indicate means (n = 3) ± S.D. **P<0.01 compared with the vehicle condition for the respective genotype.

assay reflected Sre1 activity upon low ergosterol conditions. Given that Scp1 is required for Sre1 cleavage in fission yeast [8], we also examined the effect of Scp1 deletion on ergosterol

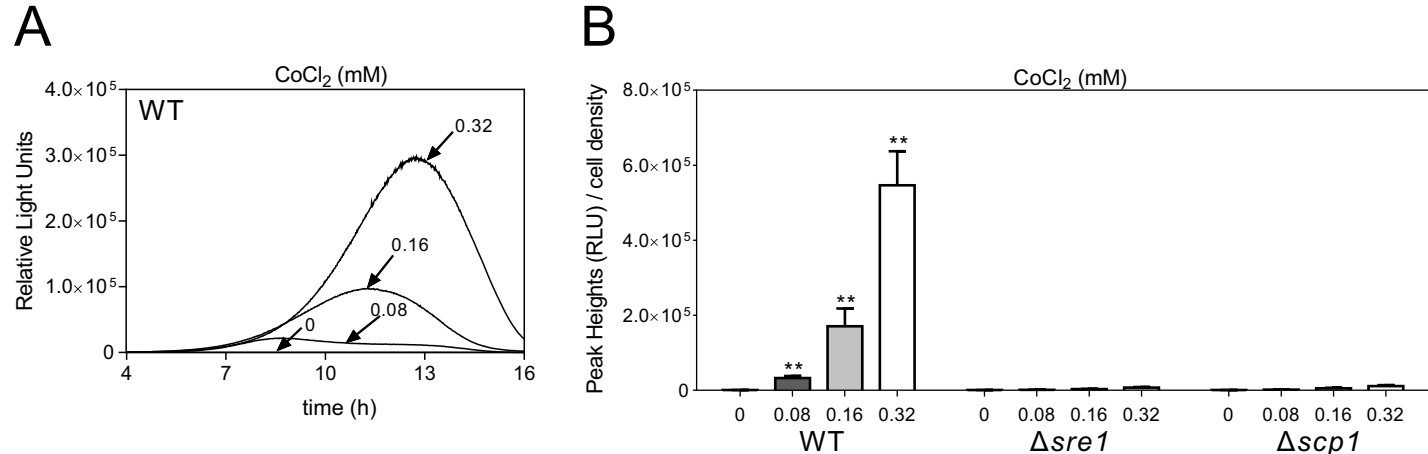

**Fig 2. CoCl₂ activates Sre1-dependent SRE2 reporter activity.** CoCl₂ (0.08 mM to 0.32 mM) induced a marked increase in Sre1 transcriptional activity. Wild-type cells harboring the multicopy reporter plasmid were cultured and assayed as described in Fig 1A–1C. (B) Two orders of magnitude reduced from wild type cells were observed in Δsre1 and Δscp1 cells. The Δsre1 and Δscp1 cells harboring the multicopy reporter plasmid were cultured and assayed as described in Fig 1A–1C. Y-axis values are the relative light units (RLU) of peak height normalized to cell density (OD660) of each sample at the peak time. The data were averaged from three independent experiments, and each sample was done in triplicate. Error bars indicate means (n = 3) ± S.D. **P<0.01 compared with the vehicle condition for the respective genotype.

### CoCl₂ activates Sre1-dependent SRE2 reporter activity

It has been known that Sre1 functions as an important oxygen sensor in fission yeast and Sre1 can be proteolytically cleaved and activated under low oxygen conditions [8]. We then tested whether hypoxia mimic Cobalt chloride (CoCl₂) could activate SRE2 reporter. As shown in Fig 2A, CoCl₂ also caused a marked dose-dependent increase in the SRE2 reporter response with a peak at about 13 hours in wild-type cells (Fig 2A). Consistently, in Δsre1 cells, the multicopy reporter response was reduced by two orders of magnitude compared to wild-type cells, indicating that CoCl₂ could activate Sre1-dependent SRE2 reporter activity (Fig 2B). Likewise, the Δscp1 mutant showed significantly lower SRE2 reporter activity (Fig 2B).

Considering that the copy number of the multicopy plasmid stably maintained cells might be affected by various factors, we constructed wild-type and Δsre1 chromosome-borne 3×SRE2::luc (R2.2) cells, respectively (CM150 and CM172 listed in Table 1). Similar to the results of cells harboring the multicopy 3×SRE2::luc (R2.2) reporter, the wild-type chromosome-borne 3×SRE2::luc (R2.2) cells could be activated by various concentrations of CLZ, TER or CoCl₂ (Fig 3A–3C) in a dose-dependent manner with a similar peak rise time. Likewise, an extremely low response upon stimulation was observed in Δsre1 chromosome-borne 3×SRE2::luc (R2.2) cells (Fig 3D–3F). These results indicate that both the episomal multicopy and the chromosome integration 3×SRE2::luc (R2.2) reporter could reflect Sre1 activity in living cells.

### Deletion of Ssp2 or/and Gsk3/Gsk31 markedly suppressed CLZ, TER or CoCl₂-induced Sre1 activity

Next, our studies focused on whether additional protein kinases were involved in regulation of Sre1 activity. Consistent with the notion that casein kinase 1 family member Hhp2 accelerates

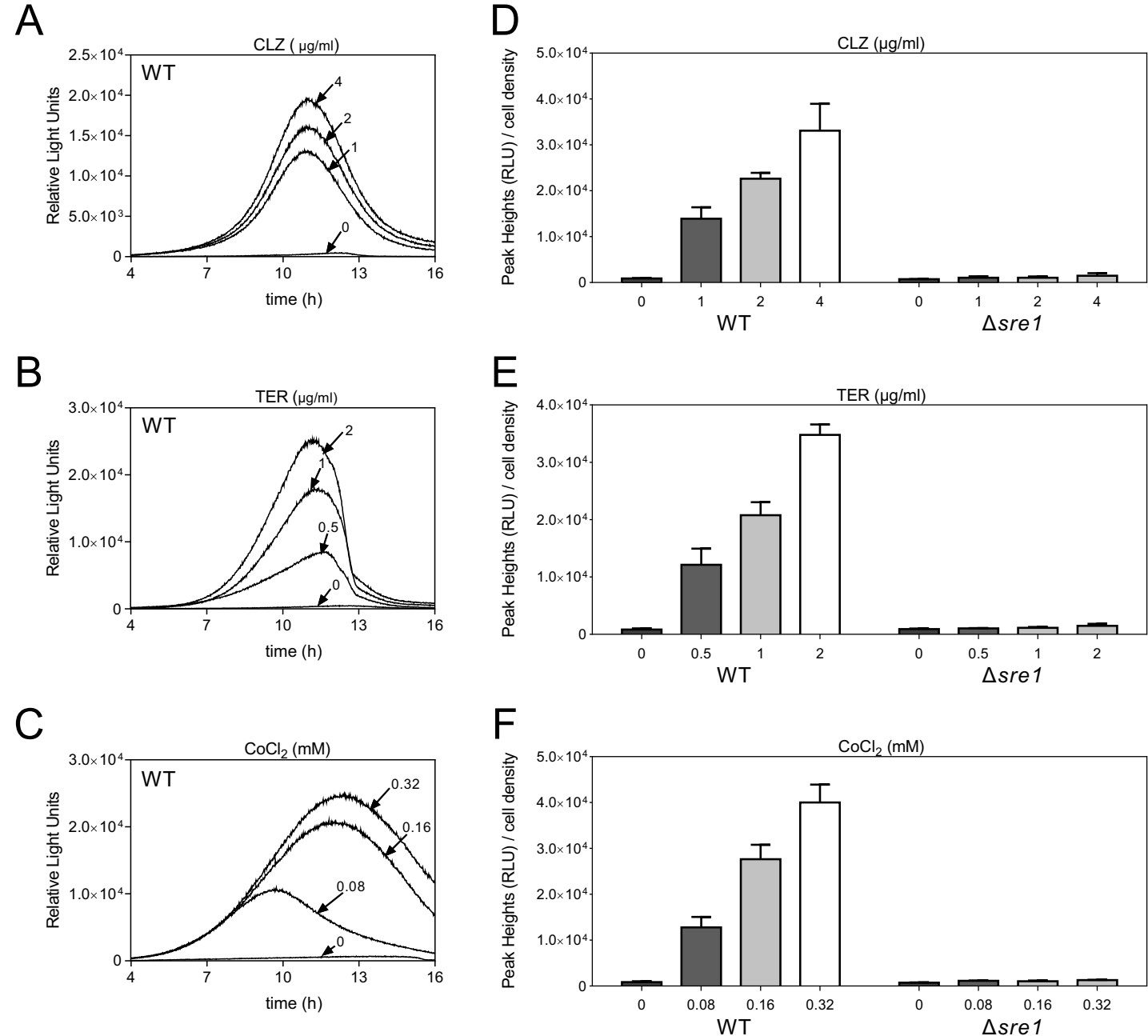

**Fig 3. The luciferase activity expressed from the chromosome-borne 3×SRE2::luc (R2.2) wild-type and Δsre1 cells.** (A) CLZ (1 μg/ml to 4 μg/ml), (B) TER (0.5 μg/ml to 2 μg/ml) and (C) CoCl₂ (0.08 mM to 0.32 mM) induced a marked increase in Sre1 transcriptional activity. Wild-type chromosome-borne 3×SRE2::luc (R2.2) cells were cultured and assayed as described in materials and methods. Y-axis values are the relative light units (RLU) of each sample. The data shown are representative of multiple experiments. (D-F) An extremely low response upon stimulation by CLZ (1 μg/ml to 4 μg/ml), TER (0.5 μg/ml to 2 μg/ml) and CoCl₂ (0.08 mM to 0.32 mM) was examined in Δsre1 chromosome-borne cells. The Δsre1 chromosome-borne cells were cultured and assayed as described in Fig 3A–3C. Y-axis values are the relative light units (RLU) of peak height normalized to the cell density (OD660) of each sample at the peak time. The data were averaged from three independent experiments, and each sample was done in triplicate. Error bars indicate means (n = 3) ± S.D.

Sre1N degradation [13], we found that deletion of *hhp2⁺* significantly increased SRE2 reporter activity in the presence/absence of CLZ, TER or CoCl₂ (Fig 4A–4C), thus further validating our reporter system. Surprisingly, in deletion of *ssp2⁺* gene, encoding the AMP-activated

A

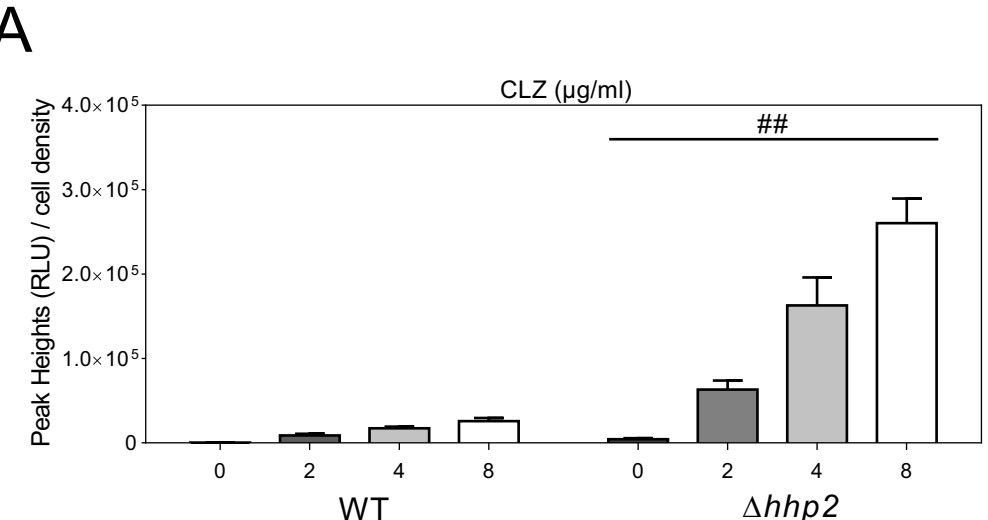

B

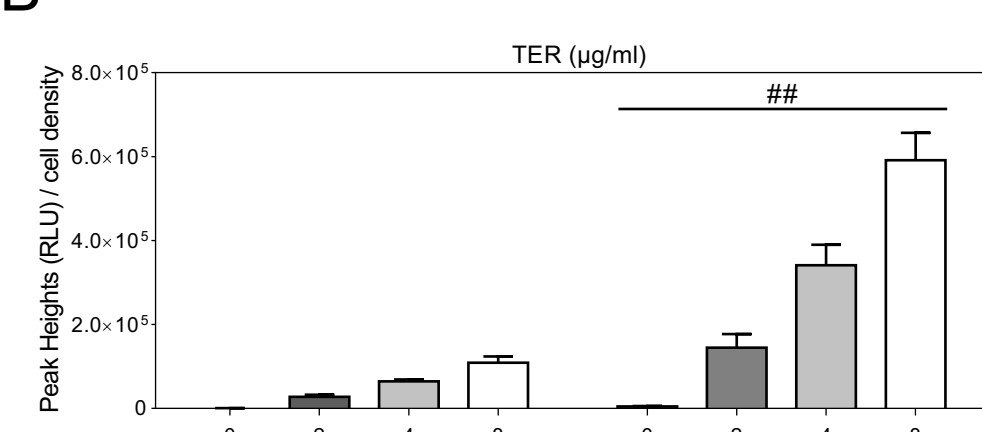

C

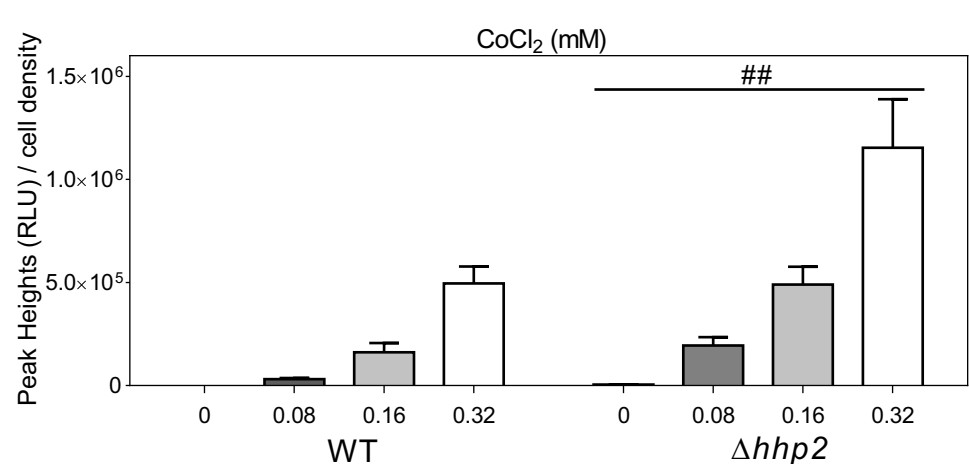

**Fig 4. Deletion of Hhp2 markedly increased Sre1 activity.** Deletion of *hhp2*⁺ gene markedly increased Sre1 activity in the presence/absence of CLZ (2 μg/ml to 8 μg/ml) (A), TER (2 μg/ml to 8 μg/ml) (B) or CoCl₂ (0.08 mM to 0.32 mM) (C). Cells harboring the multicopy reporter plasmid were cultured and assayed as described in Fig 1A–1C. Y-axis values are the relative light units (RLU) of peak height normalized to the cell density (OD660) of each sample at the peak time. The data were averaged from three independent experiments, and each sample was done in triplicate. Error bars indicate means (n = 3) ± S.D. ##P<0.01 compared with wild-type cells treated with the same drug concentration.

protein kinase (AMPK) α Subunit, CLZ, TER or CoCl₂-induced Sre1 activity was significantly suppressed (Fig 5A–5C). It should be noted that, compared with wild-type cells, deletion of Ssp2 significantly delayed the peak rising of the SRE2 reporter (Fig 5D–5F).

Previously, we identified two glycogen synthase kinases encoding genes, *gsk3*⁺ and *gsk31*⁺ as multicopysuppressors of Ssp2 deletion, and revealed a genetic interaction between Ssp2 and Gsk3 or Gsk31 in cell growth and sexual differentiation [27]. Then we sought to investigate whether Gsk3 or/and Gsk31 was required for Sre1 activity upon CLZ, TER or CoCl₂ treatment. We examined the Sre1 activity of Δ*gsk3* or Δ*gsk31* single deletion as well as Δ*gsk3*Δ*gsk31* double deletion by using SRE2 reporter. The results showed that the Δ*gsk3* single deletion or the Δ*gsk31* single deletion slightly, but the Δ*gsk3*Δ*gsk31* double deletion extremely suppressed the CLZ, TER or CoCl₂-induced Sre1 activity (Fig 5A–5C). Similar to deletion of Ssp2, the Δ*gsk3*Δ*gsk31* double deletion significantly delayed the peak rising of the SRE2 reporter (Fig 5D–5F).

We also tested the Sre1 activity in either Δ*ssp2*Δ*gsk3* or Δ*ssp2*Δ*gsk31* cells by using SRE2 reporter. Upon CLZ or TER treatment, the Sre1 activity of either Δ*ssp2*Δ*gsk3* or Δ*ssp2*Δ*gsk31* cells was slightly lower than that of Δ*ssp2* cells, but significantly lower than that of Δ*gsk3* or Δ*gsk31* cells (Fig 5A and 5B). On the other hand, upon treatment with CoCl₂, the Sre1 activity of either Δ*ssp2*Δ*gsk3* or Δ*ssp2*Δ*gsk31* was almost equal to that of Δ*ssp2* cells, slightly lower than that of Δ*gsk3* or Δ*gsk31* cells respectively (Fig 5C). Additionally, the Δ*ssp2*Δ*gsk3* or Δ*ssp2*Δ*gsk31* cells also significantly delayed the peak rising of the SRE2 reporter, similar to deletion of Ssp2 (Fig 5D–5F).

To further investigate the roles of Ssp2 and Gsk3/Gsk31 in the Sre1 activity, we constructed the strains lacking all of these three genes and tested the Sre1 activity in this triple deletion cells. As shown in Fig 5A–5C, the Δ*ssp2*Δ*gsk3*Δ*gsk31* cells showed extremely lower but still measurable Sre1 activity upon treatment with CLZ, TER or CoCl₂, compared with any of their single or double deletion cells (Fig 5A–5C). Consistently, the Δ*ssp2*Δ*gsk3*Δ*gsk31* triple deletion showed more marked temperature sensitivity than any of their single or double deletion (Fig 5G).

## Deletion of Ssp2 or/and Gsk3/Gsk31 reduced the nuclear fluorescence of GFP-Sre1N as well as GST-Sre1N protein levels

Given that Sre1N exists as a hyperphosphorylated protein that contains at least 22 phosphory-lated serine and threonine residues [8], and protein kinase Hhp2 regulates Sre1N degradation, We then wanted to know whether deletion of Ssp2, Gsk3 or Gsk31 affect Sre1N degradation. Sre1N protein tagged with GFP was visualized and the effect of deletion of Ssp2, Gsk3 or Gsk31 was examined. GFP-Sre1N is functional as its expression complemented the CoCl₂-sensitive growth defect of the Δ*sre1* cells (Fig 6A). It is known that Sre1N is released at Golgi and enters the nucleus for further promoting transcription of the target genes [28]. As expected, GFP-Sre1N clearly localized at the nucleus in the wild-type cells (Fig 6B). In Δ*gsk3* or Δ*gsk31* cells, GFP-Sre1N also localized at the nucleus, similar to that of wild-type cells. However, in Δ*ssp2*, Δ*gsk3*Δ*gsk31*, Δ*ssp2*Δ*gsk3*, Δ*ssp2*Δ*gsk31* or Δ*ssp2*Δ*gsk3*Δ*gsk31* deletion cells, significantly weaker fluorescence intensity of GFP-Sre1N was observed at the nucleus compared with that

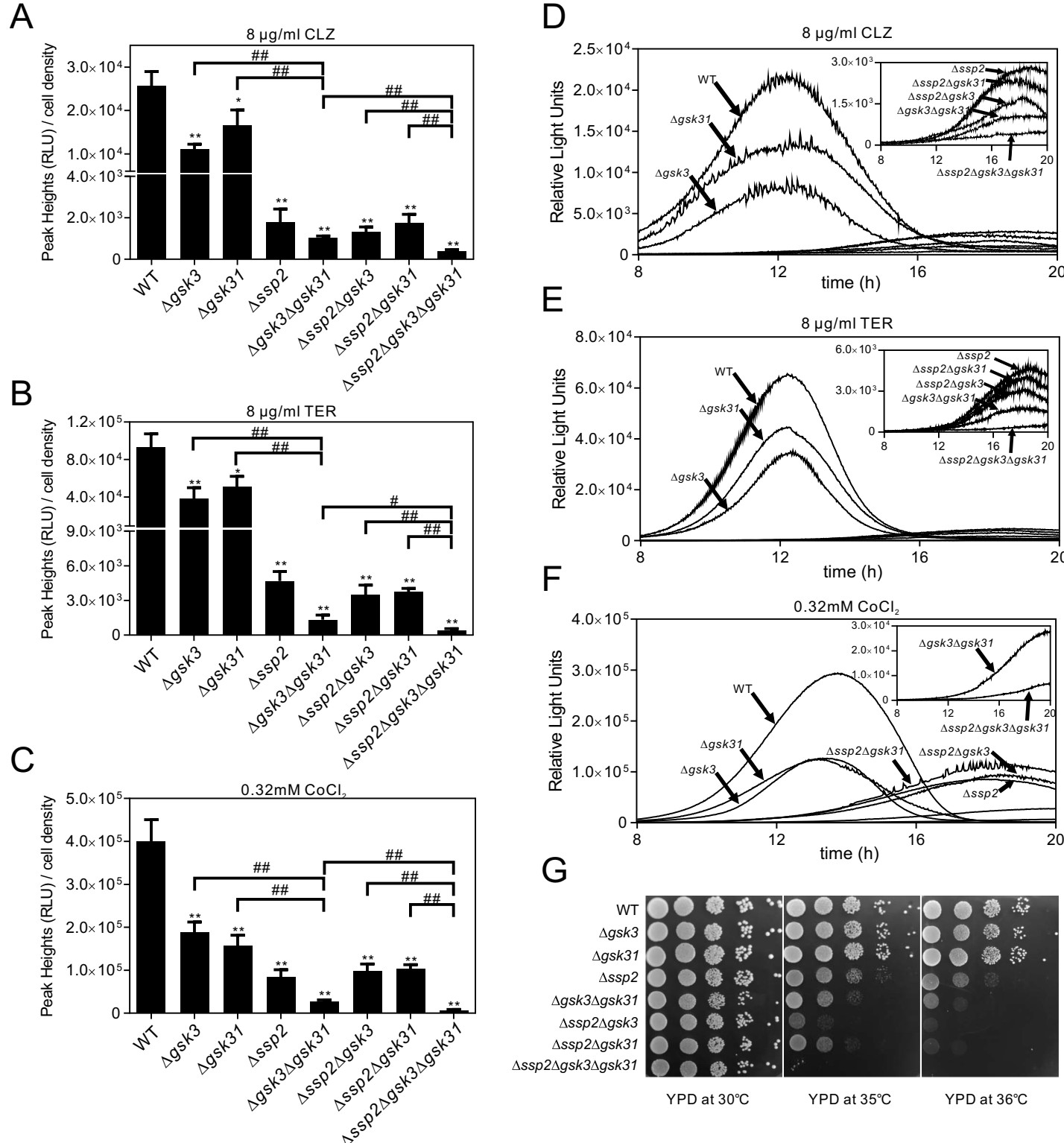

**Fig 5. Deletion of Ssp2 or/and Gsk3/Gsk31 markedly suppressed CLZ, TER or CoCl₂-induced Sre1 activity.** The Δ*ssp2*, Δ*gsk3*Δ*gsk31*, Δ*ssp2*Δ*gsk3*, Δ*ssp2*Δ*gsk31* or Δ*ssp2*Δ*gsk3*Δ*gsk31* cells showed a significant decrease in Sre1 activity and delay in the peak rising of the SRE2 reporter stimulated with 8 µg/ml CLZ (A), 8 µg/ml TER (B) or 0.32 mM CoCl₂ (C). Cells harboring the multicopy reporter plasmid were cultured and assayed as described in Fig 1A–1C. Y-axis values are the relative light units (RLU) of peak height normalized to the cell density (OD660) of each sample at the peak time. The data were averaged from three independent experiments, and each sample was done in triplicate. Error bars indicate means (n = 3) ± S.D. *P<0.05 and **P<0.01 compared with wild-type cells. #P<0.05 and ##P<0.01 compared between

different genotypes as indicated. (D-F) The data shown are representative of multiple experiments as described in Fig 5A–5C. Y-axis values are the relative light units (RLU) of each sample. (G) The Δssp2Δgsk3Δgsk31 triple deletion cells showed more marked temperature sensitivity than any of their single or double deletion. Wild-type, Δgsk3, Δgsk31, Δssp2, Δgsk3Δgsk31, Δssp2Δgsk3, Δssp2Δgsk31 or Δssp2Δgsk3Δgsk31 cells were spotted onto each plate as indicated, and then incubated at 30˚C, 35˚C or 36˚C for 3 days.

of wild-type cells (Fig 6B and 6C). As these data seem consistent with changes in Sre1N activity as well as Sre1N protein levels, we further performed the immunoblot analysis to detect the total protein levels of Sre1N in wild-type cells and the deletion mutants. GST-Sre1N is also functional as its expression complemented the CoCl₂-sensitive growth defect of the Δsre1 cells

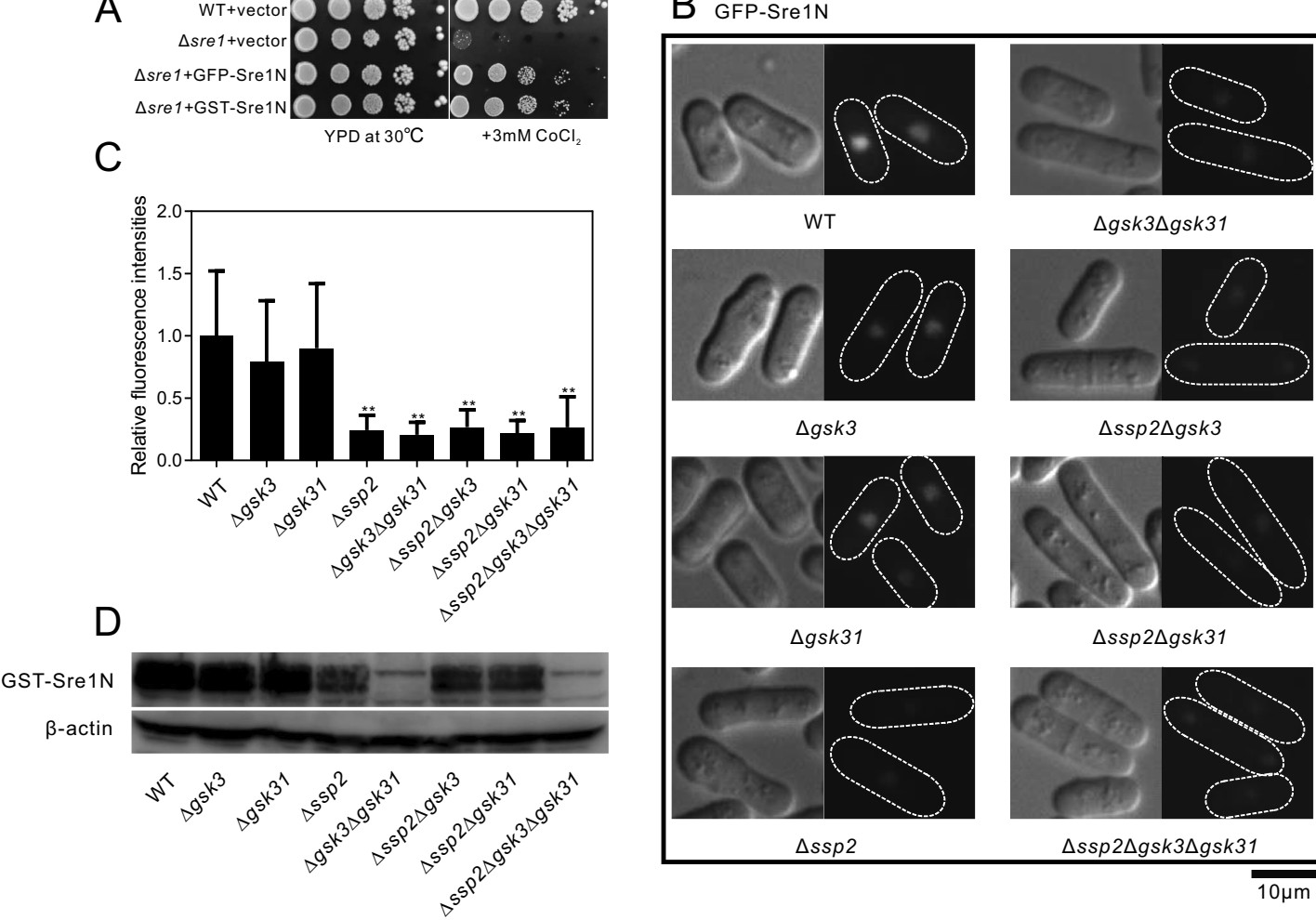

**Fig 6. Deletion of Ssp2 or/and Gsk3/Gsk31 reduced the nuclear fluorescence of GFP-Sre1N as well as GST-Sre1N protein levels.** (A) Both GFP-Sre1N and GST-Sre1N complemented the CoCl₂-sensitive growth defect of the Δsre1 cells. Cells transformed with the empty vector, the pREP1-GFP-Sre1N plasmid or pREP1-GST-Sre1N plasmid were spotted onto each plate as indicated, and then incubated at 30˚C for 4 days. (B) Significantly weaker fluorescence of GFP-Sre1N at the nucleus was observed in Δssp2, Δgsk3Δgsk31, Δssp2Δgsk3, Δssp2Δgsk31 or Δssp2Δgsk3Δgsk31 cells. Wild-type, Δgsk3, Δgsk31, Δssp2, Δgsk3Δgsk31, Δssp2Δgsk3, Δssp2Δgsk31 or Δssp2Δgsk3Δgsk31 cells expressing GFP-Sre1N were grown in EMM medium with 4 μM thiamine at 30˚C for 16 h, and then observed by fluorescence microscopy for the same 3 seconds exposure. Bar, 10 μm. (C) Fluorescence intensity quantification of GFP-tagged nucleus Sre1N. Fluorescence intensities of about 50 cells from each strain cultured at 30˚C were measured. Average values after background subtraction are shown in the bar graph. Error bars represent standard deviations. (D) The whole amount of Sre1N was markedly reduced in the Δgsk3Δgsk31 or Δssp2Δgsk3Δgsk31 cells but just a little reduced in the Δssp2, Δssp2Δgsk3 or Δssp2Δgsk31 cells as assessed by immunoblot analysis. Wild-type, Δgsk3, Δgsk31, Δssp2, Δgsk3Δgsk31, Δssp2Δgsk3, Δssp2Δgsk31 or Δssp2Δgsk3Δgsk31 cells were transformed with pREP1-GST-Sre1N and were cultured in liquid EMM to mid-log phase. Total cell lysates were prepared as described in materials and methods, and then the protein extracts were subjected to SDS-PAGE, and immunoblotted using anti-GST antibodies. β-actin was used as a loading control.

(Fig 6A). Consistent with their weak fluorescence intensity, the whole amount of GST-Sre1N was reduced in the mutants except that in Δ*gsk3* or Δ*gsk31* cells (Fig 6D). However, the total amount of GST-Sre1N was markedly different between Δ*ssp2* and Δ*gsk3*Δ*gsk31* deletion cells. The immunoblot showed a dramatic decrease in Sre1N levels in the Δ*gsk3*Δ*gsk31* double mutant cells, while deletion of Ssp2 has just a minor effect on Sre1N levels (Fig 6D). This was different from the nuclear localization, where Δ*ssp2* mutant had a major effect that was similar to the Δ*gsk3*Δ*gsk31* double mutant (Fig 6B and 6C). In comparing the Δ*gsk3*Δ*gsk31* double mutant versus the Δ*ssp2*Δ*gsk3*Δ*gsk31* triple mutant, there is no additive defect in Sre1N levels and the two strains look identical (Fig 6D). These results suggest that Gsk3/Gsk31 might play the primary role in regulating Sre1N degradation, whereas Ssp2 might regulate not only Sre1N degradation but also nuclear localization of Sre1N.

## Discussion

In fission yeast, Sre1, the homologue of mammalian SREBP, regulates sterol homeostasis and hypoxia adaptation [29]. It was known that, as a negative regulator of Sre1, casein kinase 1 family member Hhp2 accelerates Sre1N degradation. However, studies on additional kinases involved in Sre1 activity regulation are still limited. Here, we identified AMPKα Subunit Ssp2 and Glycogen Synthase Kinases Gsk3/Gsk31 as positive regulators of Sre1, which are involved in regulation of Sre1 activity via inhibiting degradation and accelerating translocation of Sre1N into the nucleus. To our knowledge, this is the first report to reveal a novel requirement for protein kinases Ssp2 and Gsk3/Gsk31 in regulation of Sre1 activity in fission yeast.

Three evidences support that our luciferase reporter system could reflect Sre1 activity. First, ergosterol biosynthesis inhibitors, namely CLZ, TER and FEN could induce a marked increase in transcriptional activity of Sre1 in a dose-dependent manner. Second, ergosterol biosynthesis inhibitors-induced Sre1 activity was abolished in Δ*sre1* or Δ*scp1* cells. Third, loss of Hhp2, a negative regulator of Sre1, significantly increased transcriptional activity of Sre1 in the presence/absence of ergosterol biosynthesis inhibitors, such as CLZ or TER.

Our previous studies found that Gsk3 and Gsk31 function redundantly in cell growth at restrictive temperatures and sexual differentiation [27]. In present study, several lines of evidence support the hypothesis that Gsk3 and Gsk31 function redundantly in regulation of Sre1 activity, as well as Ssp2 and Gsk3/31 act on parallel in regulation of Sre1 activity. First, CLZ, TER or $CoCl_2$-induced Sre1 activity in Δ*gsk3*Δ*gsk31* cells was significantly reduced compared to wild-type cells, but slightly reduced in Δ*gsk3* or Δ*gsk31* cells. Second, CLZ, TER or $CoCl_2$-induced Sre1 activity in Δ*ssp2*Δ*gsk3* or Δ*ssp2*Δ*gsk31* was only slightly lower than or almost equal to Δ*ssp2* cells. Third, the deletion of *ssp2*[+], *gsk3*[+] and *gsk31*[+], *ssp2*[+] and *gsk3*[+], or *ssp2*[+] and *gsk31*[+] significantly delayed the peak rising of the SRE2 reporter, but the deletion of *gsk3*[+] or *gsk31*[+] did not. Forth, the Δ*ssp2*Δ*gsk3*Δ*gsk31* cells showed the lowest Sre1 activity compared to any of their single or double deletions. These results suggested that there is a genetic interaction between Ssp2 and Gsk3/Gsk31, and Ssp2 and Gsk3/Gsk31 may act on parallel pathway in regulation of Sre1 activity.

Furthermore, we found that the fluorescence of GFP-Sre1N at the nucleus observed in Δ*ssp2*, Δ*gsk3*Δ*gsk31*, Δ*ssp2*Δ*gsk3*, Δ*ssp2*Δ*gsk31* or Δ*ssp2*Δ*gsk3*Δ*gsk31* cells was significantly weakened compared with that in wild-type cells. To our surprises, while nuclear accumulation of GFP-Sre1N appeared to be diminished to a similar extent in the Δ*ssp2* mutant and the Δ*gsk3*Δ*gsk31* double mutant, the total amount of GST-Sre1N was markedly different in these two mutants. Thus, it seems possible that Ssp2 regulates not only the degradation of Sre1N but also its translocation to the nucleus, whereas Gsk3/Gsk31 regulate mainly its degradation. Since casein kinase 1 family member Hhp2 accelerates Sre1N degradation, our results

suggested that Ssp2/Gsk3/Gsk31 might act as inhibitors of Hhp2, or alternatively act on Sre1N activity independently of Hhp2. Previous studies suggested that Sre1 cleavage, Sre1N stability and Sre1N DNA binding are involved in the regulation of Sre1 activity [30–32]. Based on our present results, we propose that Ssp2 and Gsk3/Gsk31 might affect Sre1N stability to regulate Sre1 activity. However, it is currently undetermined whether Ssp2 and Gsk3/31 are involved in Sre1 cleavage or Sre1N DNA binding for regulation of Sre1 activity.

In conclusion, our findings establish new functional link between Sre1 and three protein kinases, namely Ssp2, Gsk3 and Gsk31. The present data strongly suggest that Ssp2 and Gsk3/Gsk31 play cooperative but distinct roles in the regulation of Sre1 activity in fission yeast. Understanding whether Ssp2 or Gsk3/Gsk31 directly phosphorylates Sre1N and inhibits its degradation is important questions to be addressed in the future.

## Supporting information

**S1 Appendix. The original uncropped and unadjusted western blotting images and all individual data points within curve and column graphs.**
(ZIP)

## Author Contributions

**Conceptualization:** Takayoshi Kuno, Yue Fang.

**Data curation:** Hao Miao, Qiannan Liu, Guanglie Jiang, Wen Zhang, Kun Liu, Xiang Gao, Yujie Huo, Si Chen.

**Formal analysis:** Hao Miao, Qiannan Liu, Guanglie Jiang, Wen Zhang, Kun Liu, Xiang Gao, Yujie Huo, Si Chen.

**Funding acquisition:** Si Chen, Yue Fang.

**Investigation:** Hao Miao, Qiannan Liu, Guanglie Jiang, Wen Zhang, Kun Liu, Xiang Gao, Yujie Huo, Si Chen.

**Methodology:** Toshiaki Kato, Norihiro Sakamoto, Takayoshi Kuno, Yue Fang.

**Project administration:** Yue Fang.

**Resources:** Takayoshi Kuno, Yue Fang.

**Software:** Hao Miao, Qiannan Liu, Guanglie Jiang, Wen Zhang, Xiang Gao.

**Supervision:** Yue Fang.

**Validation:** Hao Miao, Qiannan Liu, Guanglie Jiang, Wen Zhang, Kun Liu, Xiang Gao, Si Chen.

**Visualization:** Hao Miao, Yue Fang.

**Writing – original draft:** Hao Miao, Yue Fang.

**Writing – review & editing:** Yue Fang.

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
