## [Decision Letter · Decision Letter 0]

20 Nov 2019

PONE-D-19-30640

AMPKα Subunit Ssp2 and Glycogen Synthase Kinases Gsk3/Gsk31 are involved in regulation of sterol regulatory element-binding protein (SREBP) activity in fission yeast

PLOS ONE

Dear Dr Fang,

Thank you for submitting your manuscript to PLOS ONE. After careful consideration, we feel that it has merit but does not fully meet PLOS ONE’s publication criteria as it currently stands. Therefore, we invite you to submit a revised version of the manuscript that addresses the points raised during the review process.

Your manuscript was evaluated by two experts in the field and their reports were returned. As you see, both referees gave favorable reviews. However, they also raised several points, which I agree with. Both referees pointed out caveats of the episomal reporter assay. Please reply satisfactorily to their comments possibly by adding new data. They also raised another point with regards to the levels of nuclear and cytoplasmic GFP-Sre1N. Please address this important point. In addition, Referee 2 raised a few concerns. I hope that you could revise the manuscript in response to these comments. Then, I would be happy to consider acceptance of this manuscript.

We would appreciate receiving your revised manuscript by Dr Fang. To enhance the reproducibility of your results, we recommend that if applicable you deposit your laboratory protocols in protocols.io, where a protocol can be assigned its own identifier (DOI) such that it can be cited independently in the future. For instructions see: http://journals.plos.org/plosone/s/submission-guidelines#loc-laboratory-protocols

We look forward to receiving your revised manuscript.

Kind regards,

Takashi Toda PhD

Academic Editor

PLOS ONE

Journal Requirements:

Reviewers' comments:

Reviewer's Responses to Questions

**Comments to the Author**

1. Is the manuscript technically sound, and do the data support the conclusions?

Reviewer #1: Partly

Reviewer #2: Partly

2. Has the statistical analysis been performed appropriately and rigorously? 

Reviewer #1: Yes

Reviewer #2: Yes

3. Have the authors made all data underlying the findings in their manuscript fully available?

Reviewer #1: Yes

Reviewer #2: Yes

4. Is the manuscript presented in an intelligible fashion and written in standard English?

Reviewer #1: Yes

Reviewer #2: Yes

5. Review Comments to the Author

Reviewer #1: SREBP (Sre1 in fission yeast) is an evolutionally conserved transcriptional factor regulating the expression of genes involved in sterol biogenesis. In this manuscript, Yue Fang et al develop the reporter system to measure the level of SREBP-dependent transcription in fission yeast. The authors show that the expression of the reporter is upregulated in cells treated with inhibitors of sterol biosynthesis or CoCl2, and that deletion of genes encoding AMPK (Ssp2) and/or GSK3s (Gsk3 and Gsk31) greatly diminishes the expression of the reporter. The authors find the intensity of the GFP-Sre1N fluorescence in the nucleus is reduced in mutant cells lacking ssp2 and/or gsk3/gsk31 genes, and speculate that AMPK and GSK may synergistically inhibit degradation of the N-terminal part of Sre1, which is released by protein cleavage on the Golgi, translocated to the nucleus and promotes the transcription of target genes.

While I think that study is well-conducted and the presented results are largely clear, I have two major concerns described below:

1) In Figure 4, they measure the luciferase activity expressed form the episomally-introduced multicopy plasmid. Although the authors conclude that the reduction of the luciferase activity in the mutant cells lacking ssp2 and/or gsk3/gsk31 is caused by deficiency of Sre1 (SREBP) function, it may be possibly caused by the reduction of the plasmid copy number. Considering that the copy number of the plasmid stably maintained cells is greatly affected by various factors, the authors cannot exclude the possibility that deletion of these genes may somehow reduce the plasmid copy number as long as the episomal plasmid is used. I strongly suggest the authors to perform the experiment using cells in which the reporter construct is integrated to the chromosome. At the minimum, they should confirm that the cells used in the experiment harbor the same number of the reporter plasmid by measuring the amount of the plasmid DNA in the cells.

2) In Figure 5, the authors clams that the reduction of GFP-Sre1N fluorescence in the nucleus in the mutant cells is caused by degradation of the GFP-Sre1N protein. However, I believe that it is equally possible that deletion of ssp2 and/or gsk3/gsk31 genes may somehow perturb translocation of the GFP-Sre1N protein from the cytoplasm to the nucleus and the protein may be dispersed throughout the cells. The authors should perform the immunoblotting experiment to show more directly that the whole amount of GFP-Sre1N is reduced in the mutants. If the accumulation of degraded product of GFP-Sre1N in the mutant could be detected in immunoblot, their conclusion would be further strengthened. If phosphorylation of Sre1N causes its band-shift, the author may be also able to see how much the Sre1N phosphorylation is affected by deletion of ssp2 and/or gsk3/gsk31 genes.

Reviewer #2: This paper by Miao et al. investigates protein kinases that regulate the SREBP response in fission yeast. Past work has shown that CK1 kinase Hhp2 phosphorylates Sre1N (the cleaved and activated region of SREBP) to accelerate its degradation. Here, the authors generated a luciferase-based reporter for Sre1N activity, and identified a role for AMPK/Ssp2 and GSK-3 orthologs Gsk3 and Gsk31 in regulating Sre1N activity. The authors’ data support a model where Ssp2, Gsk3, and Gsk31 promote Sre1N activity and/or levels. The underlying mechanisms remain undefined in the current work, but identifying these new regulators would represent progress on this pathway and therefore would be of interest to the field. I have several technical and interpretation comments/concerns that should be addressed before publishing this work.

Technical:

1. The luciferase-based reporter is introduced into cells as a plasmid, but I do not see how maintenance of the plasmid is selected. The Methods section states that transformants were cultured in ‘normal EMM media.’ Is there an auxotrophic or antibiotic-based selection to make sure that cells have and maintain the plasmid? This is important because changes in plasmid maintenance would alter the results.

2. For the luciferase-based assays, is the total luciferase signal normalized to cell number? It is important to control for changes in the growth rate of cells, so the light units should be normalized to cell number in each case.

Data interpretation:

1. sre1∆ cells retain some luciferase activity upon treatment with CoCl2, suggesting Sre1-independent transcription of the reporter. I do not disagree with this interpretation, but the authors should note that the response is two orders of magnitude reduced from wild type cells.

2. The authors measure Sre1N-GFP levels in the nucleus, and conclude that degradation has been impacted in their mutants. However, their measured differences are also consistent with altered shuttling between the nucleus and cytoplasm. They should measure total cellular levels if they want to conclude changes in total protein levels.

3. The authors could note that their results are consistent with Ssp2/Gsk3/Gsk31 acting as inhibitors of Hhp2, or alternatively acting on Sre1N activity independently of Hhp2.

4. The authors’ data seem consistent with changes in Sre1N activity as well as Sre1N protein levels. They might add this possibility into the text. In particular, they have not provided data for changes in Sre1N degradation in the new kinase mutants (e.g. lines 297-299). To make this conclusion, they would have to measure degradation rates with different assays.

6. PLOS authors have the option to publish the peer review history of their article (what does this mean?). If published, this will include your full peer review and any attached files.

Reviewer #1: Yes: Shigeaki Saitoh

Reviewer #2: No

---

## [Author Response · Author response to Decision Letter 0]

8 Jan 2020

January 04, 2020

PLoS ONE

Dear Dr. Takashi Toda:

Thank you for your e-mail of November 21, 2019 on our manuscript entitled “AMPKα Subunit Ssp2 and Glycogen Synthase Kinases Gsk3/Gsk31 are involved in regulation of sterol regulatory element-binding protein (SREBP) activity in fission yeast” by Hao Miao et al. (PONE-S-19-38064). We would like to thank you for your very helpful comments and suggestions, and for inviting us to revise our manuscript.

In your e-mail, you’ve suggested that our revisions should address several specific points raised by two reviewers as mentioned below. Consequently, we performed a number of experiments based on the comments and were successful in obtaining new data. Thus, we address each point raised by the two reviewers.

A list of the revisions which addresses reviewers' comments is as follows:

Reviewer #1

Comments 

1. In Figure 4, they measure the luciferase activity expressed form the episomally-introduced multicopy plasmid. Although the authors conclude that the reduction of the luciferase activity in the mutant cells lacking ssp2 and/or gsk3/gsk31 is caused by deficiency of Sre1 (SREBP) function, it may be possibly caused by the reduction of the plasmid copy number. Considering that the copy number of the plasmid stably maintained cells is greatly affected by various factors, the authors cannot exclude the possibility that deletion of these genes may somehow reduce the plasmid copy number as long as the episomal plasmid is used. I strongly suggest the authors to perform the experiment using cells in which the reporter construct is integrated to the chromosome. At the minimum, they should confirm that the cells used in the experiment harbor the same number of the reporter plasmid by measuring the amount of the plasmid DNA in the cells. 

Response:

Accordingly, we constructed wild-type and Δsre1 chromosome-borne 3×SRE2::luc (R2.2) strains named CM150 and CM172, respectively (page 6, lines 127-131 / page 7, lines 150-153 / listed in table.1). The results showed that wild-type chromosome-borne 3×SRE2::luc (R2.2) cells could be activated by various concentration of CLZ, TER or CoCl2 (Fig 3A-C) in a dose-dependent manner, but in Δsre1 chromosome-borne 3×SRE2::luc (R2.2) cells, an extremely low response upon stimulation was observed (Fig 3D-F). These results are consistent with those obtained with multicopy reporter and suggested that both the episomal multicopy and the chromosome integration 3×SRE2::luc (R2.2) reporter could reflect Sre1 activity in living cells. These results are incorporated in the revised manuscript (page 12-13, lines 259-269 / Fig 3 and legend, page22, lines 449-462). 

2. In Figure 5, the authors claims that the reduction of GFP-Sre1N fluorescence in the nucleus in the mutant cells is caused by degradation of the GFP-Sre1N protein. However, I believe that it is equally possible that deletion of ssp2 and/or gsk3/gsk31 genes may somehow perturb translocation of the GFP-Sre1N protein from the cytoplasm to the nucleus and the protein may be dispersed throughout the cells. The authors should perform the immunoblotting experiment to show more directly that the whole amount of GFP-Sre1N is reduced in the mutants. If the accumulation of degraded product of GFP-Sre1N in the mutant could be detected in immunoblot, their conclusion would be further strengthened. If phosphorylation of Sre1N causes its band-shift, the author may be also able to see how much the Sre1N phosphorylation is affected by deletion of ssp2 and/or gsk3/gsk31 genes.

Response:

We thank the reviewer for giving us this very helpful comment. Accordingly, we performed the immunoblot assay to detect the total Sre1N protein levels in wild-type and mutant cells (page 10, lines 201-214). Consistent with fluorescence intensity in wild-type cells and the mutants, the whole amount of Sre1N was markedly reduced in these mutants compared with that of wild-type cells. Notably, the whole amount of Sre1N in Δssp2Δgsk3Δgsk31 triple deletion was lower than that in any of their single or double deletions assessed by immunoblot analysis (Fig 6D). These are incorporated in the revised manuscript (page 15-16, lines 313-314, 329-337 / Fig 6D and legend, page 24-25, lines 493-494, 507-514).

Reviewer #2

Comments 

Technical:

1. The luciferase-based reporter is introduced into cells as a plasmid, but I do not see how maintenance of the plasmid is selected. The Methods section states that transformants were cultured in ‘normal EMM media.’ Is there an auxotrophic or antibiotic-based selection to make sure that cells have andmaintain the plasmid? This is important because changes in plasmid maintenance would alter the results.

Response:

Accordingly, there is an auxotrophic based selection for the transformants to make sure that cells have and maintain the plasmid described as follow: the multicopy 3×SRE2::luc (R2.2) reporter vector containing leucine marker can complement the S.pombe mutations leu1 (for example, wide-type cells HM123 (h- leu1-32)), then the transformants would be successfully selected in EMM media without leucine. These are incorporated in the revised manuscript (page 7-8, lines 153-156).

2. For the luciferase-based assays, is the total luciferase signal normalized to cell number? It is important to control for changes in the growth rate of cells, so the light units should be normalized to cell number in each case.

Response:

Accordingly, for all the luciferase-based assays, we measured the cell density (OD660) of each groups at the peak time, and the luciferase light units of peak height were normalized to the corresponding cell density at the peak time. The figures and figure legends (Fig 1D-F and legend, page 21, lines 431-433 / Fig 2B and legend, page 22, lines 443-444 / Fig 3D-F and legend, page 22, lines 459-460 / Fig 4A-C and legend, page 23, lines 468-469 / Fig 5A-C and legend, page 23, lines 480-481) as well as a description of the results (page 14, lines 297-299) are modified in the revised manuscript.

Data interpretation:

1. sre1∆ cells retain some luciferase activity upon treatment with CoCl2, suggesting Sre1-independent transcription of the reporter. I do not disagree with this interpretation, but the authors should note that the response is two orders of magnitude reduced from wild type cells.

Response:

Although ∆sre1 cells transformed with 3×SRE2::luc (R2.2) multicopy plasmid reporter retain some luciferase activity upon treatment with CoCl2, it is still unclear what its physiological significance is. We further constructed wild-type and Δsre1 chromosome-borne 3×SRE2::luc (R2.2) strains, and found that CoCl2-induced increase in SRE2 reporter activity was almost abolished in Δsre1 cells (Fig 3). Therefore, we removed the related description of Sre1-independent activation as well as partially enlarged part in Fig 1 and Fig 2. According to the reviewer’s comment, we noted that, when sre1+ was knocked out, Sre1 activity stimulated with CoCl2 decreased to less than 1% compared to that of wild-type cells. These are included in the revised manuscript (page 12, line 248, 254-256 / Fig 1D-F / Fig 2B and legend, page 21, line 437, 440-441).

2. The authors measure Sre1N-GFP levels in the nucleus, and conclude that degradation has been impacted in their mutants. However, their measured differences are also consistent with altered shuttling between the nucleus and cytoplasm. They should measure total cellular levels if they want to conclude changes in total protein levels.

Response:

Accordingly, immunoblot analysis was performed to detect the total Sre1N protein levels in wild-type cells and the mutants (page 10, lines 201-214). Consistent with their decreased fluorescence intensity of GFP-Sre1N in the nucleus, the total cellular protein levels of GST-Sre1N was also markedly reduced in the mutants. These results were included in the revised manuscript (page 15-16, lines 313-314, 329-337 / Fig 6D and legend, page 24-25, lines 493-494, 507-514).

3. The authors could note that their results are consistent with Ssp2/Gsk3/Gsk31 acting as inhibitors of Hhp2, or alternatively acting on Sre1N activity independently of Hhp2.

Response:

We thank this reviewer for giving us these helpful comments. Accordingly, we added some comments on these issues. These are incorporated in the revised manuscript (page 18, lines 380-382).

4. The authors’ data seem consistent with changes in Sre1N activity as well as Sre1N protein levels. They might add this possibility into the text. In particular, they have not provided data for changes in Sre1N degradation in the new kinase mutants (e.g. lines 297-299). To make this conclusion, they would have to measure degradation rates with different assays.

Response:

Accordingly, we added the possibility that our data are consistent with changes in Sre1N activity in the text (page15, line 328-329). To test whether the changes in Sre1N activity were consistent with Sre1N protein levels in the kinase mutants, we performed the immunoblot analysis to detect the total protein levels of Sre1N in wild-type cells and the deletion mutants. The results showed that the whole amount of GST-Sre1N was markedly reduced in the mutants except that in Δgsk3 or Δgsk31 cells. Notably, similar to the changes in Sre1N activity detected by luciferase reporter assay, the whole amount of Sre1N in Δssp2Δgsk3Δgsk31 triple deletion was lower than that in any of their single or double deletions. These results suggested that the changes in Sre1N activity were consistent with Sre1N protein levels. These are incorporated in the revised manuscript (page17-18, lines 374-380)

We hope that these revisions would satisfy your requirements. Thank you very much for your time and consideration.

Sincerely,

Yue Fang, M.D, Ph.D.

Department of Microbial and Biochemical Pharmacy,

School of Pharmacy, China Medical University,

No.77 Puhe Road, Shenyang North New Area, Shenyang 110112, China

TEL: +86-18900910820, FAX: +86-24-31939448 

Email: yfang@cmu.edu.cn

---

## [Decision Letter · Decision Letter 1]

22 Jan 2020

PONE-D-19-30640R1

AMPKα Subunit Ssp2 and Glycogen Synthase Kinases Gsk3/Gsk31 are involved in regulation of sterol regulatory element-binding protein (SREBP) activity in fission yeast

PLOS ONE

Dear Dr Fang,

Thank you for submitting your manuscript to PLOS ONE. After careful consideration, we feel that it has merit but does not fully meet PLOS ONE’s publication criteria as it currently stands. Therefore, we invite you to submit a revised version of the manuscript that addresses the points raised during the review process.

ACADEMIC EDITOR: 

Dr Fang,

Thank you for submitting the revised manuscript, which was now evaluated by the two original referees.

         Both referees acknowledged the revision; however, they raised the same point, that is the interpretation of the immunoblotting data shown in Figure 6D. They pointed out the possibility that Ssp2 and Gsk31/32 may regulate GFP-Sre1N in a distinct manner; Ssp2 mainly regulates its translocation (nuclear import/retention), while Gsk31/32 regulate overall protein levels. Having seen the data myself, I agree with these referees’ point. Therefore, I strongly encourage you rephrasing/adding some discussion which incorporates this interesting point.

I am looking forward to receiving your new revised manuscript. 

We would appreciate receiving your revised manuscript in 30 days. To enhance the reproducibility of your results, we recommend that if applicable you deposit your laboratory protocols in protocols.io, where a protocol can be assigned its own identifier (DOI) such that it can be cited independently in the future. For instructions see: http://journals.plos.org/plosone/s/submission-guidelines#loc-laboratory-protocols

We look forward to receiving your revised manuscript.

Kind regards,

Takashi Toda PhD

Academic Editor

PLOS ONE

Reviewers' comments:

Reviewer's Responses to Questions

**Comments to the Author**

1. If the authors have adequately addressed your comments raised in a previous round of review and you feel that this manuscript is now acceptable for publication, you may indicate that here to bypass the “Comments to the Author” section, enter your conflict of interest statement in the “Confidential to Editor” section, and submit your "Accept" recommendation.

Reviewer #1: (No Response)

Reviewer #2: (No Response)

2. Is the manuscript technically sound, and do the data support the conclusions?

Reviewer #1: Yes

Reviewer #2: Partly

3. Has the statistical analysis been performed appropriately and rigorously? 

Reviewer #1: Yes

Reviewer #2: Yes

4. Have the authors made all data underlying the findings in their manuscript fully available?

Reviewer #1: Yes

Reviewer #2: Yes

5. Is the manuscript presented in an intelligible fashion and written in standard English?

Reviewer #1: Yes

Reviewer #2: Yes

6. Review Comments to the Author

Reviewer #1: The criticisms raised against the original version are properly responded, and the manuscript is satisfactorily revised. I have one minor comment regarding the interpretation of the new result of immunoblotting in Figure 6.

Minor point:

While nuclear accumulation of GFP-Sre1N appears to be diminished to a similar extent in the ssp2 mutant and the gsk3 gsk31 double mutant, the total amount of GST-Sre1N was markedly different in these two mutants (Figure 6B and D). Thus, it seems possible that Ssp2 regulates not only the degradation of Sre1N but also its translocation to the nucleus, whereas Gsks regulate mainly its degradation. It may be better to discuss this possibility.

Reviewer #2: The authors have done a nice job revising the manuscript and adding new data to address the reviewer comments. I have one lingering concern related to the new immnunoblot experiment in Figure 6D. This new experiment was added in response to both reviewers, who noted that the authors should test total cellular levels of GST-Sre1N. The authors conclude that levels are decreased in a similar pattern to the nuclear levels that were determined by microscopy (e.g. page 16, lines 335-337; and stated again in the Discussion). I disagree and suggest changing the text to better reflect the actual results. The immunoblot shows a dramatic decrease in Sre1N levels in the gsk3 gsk31 double mutant cells. Deletion of ssp2 has a minor (if any) effect on these levels, and the authors would need to quantify a better exposure to make this conclusion. This is different from the nuclear localization, where ssp2 mutant had a major effect that was similar to the gsk3 gsk31 double mutant. In comparing the gsk3 gsk31 double mutant versus the ssp2 gsk3 gsk31 triple mutant, there is no additive defect in Sre1N levels and the two strains look identical. These results suggest that Gsk3 and Gsk31 play the primary role in regulating Sre1N protein levels, whereas Ssp2 might regulate nuclear localization of Sre1N. The authors can address this comment by editing the text without additional experiments. The fact that this immunoblot does not perfectly repeat the microscopy experiment is actually quite interesting and reveals the potential for some specific roles for Ssp2 versus Gsk3/31 in regulating Sre1N. I encourage the authors to revise their interpretation to better reflect this nice result.

7. PLOS authors have the option to publish the peer review history of their article (what does this mean?). If published, this will include your full peer review and any attached files.

Reviewer #1: Yes: Shigeaki Saitoh

Reviewer #2: No

---

## [Author Response · Author response to Decision Letter 1]

23 Jan 2020

January 24, 2020

PLoS ONE

Dear Dr. Takashi Toda:

Thank you for your e-mail of January 23, 2020 on our manuscript entitled “AMPKα Subunit Ssp2 and Glycogen Synthase Kinases Gsk3/Gsk31 are involved in regulation of sterol regulatory element-binding protein (SREBP) activity in fission yeast” by Hao Miao et al. (PONE-D-19-30640R1). We would like to thank you for your very helpful comments and suggestions, and for inviting us to revise our manuscript. Both reviewers raised the same point that is the interpretation of the immunoblotting data shown in Figure 6D. Accordingly, we revised our manuscript.

The revision which addresses reviewer's comment is as follows:

Reviewer #1

Comment 

The criticisms raised against the original version are properly responded, and the manuscript is satisfactorily revised. I have one minor comment regarding the interpretation of the new result of immunoblotting in Figure 6.

Minor point:

While nuclear accumulation of GFP-Sre1N appears to be diminished to a similar extent in the ssp2 mutant and the gsk3 gsk31 double mutant, the total amount of GST-Sre1N was markedly different in these two mutants (Figure 6B and D). Thus, it seems possible that Ssp2 regulates not only the degradation of Sre1N but also its translocation to the nucleus, whereas Gsks regulate mainly its degradation. It may be better to discuss this possibility. 

Response:

Accordingly, we added some comments on this issue in the revised manuscript (pages 3, lines 46-50 / page 5, lines 99-101 / page 15, line 310 / page 16, lines 334-344 / page 17, lines 353-354 / page 18, lines 379-384, 395 / page 25, lines 512-515).

Reviewer #2

Comment 

The authors have done a nice job revising the manuscript and adding new data to address the reviewer comments. I have one lingering concern related to the new immnunoblot experiment in Figure 6D. This new experiment was added in response to both reviewers, who noted that the authors should test total cellular levels of GST-Sre1N. The authors conclude that levels are decreased in a similar pattern to the nuclear levels that were determined by microscopy (e.g. page 16, lines 335-337; and stated again in the Discussion). I disagree and suggest changing the text to better reflect the actual results. The immunoblot shows a dramatic decrease in Sre1N levels in the gsk3 gsk31 double mutant cells. Deletion of ssp2 has a minor (if any) effect on these levels, andthe authors would need to quantify a better exposure to make this conclusion. This is different from the nuclear localization, where ssp2 mutant had a major effect that was similar to the gsk3 gsk31 double mutant. In comparing the gsk3 gsk31 double mutant versus the ssp2 gsk3 gsk31 triple mutant, there is no additive defect in Sre1N levels and the two strains look identical. These results suggest that Gsk3 and Gsk31 play the primary role in regulating Sre1N protein levels, whereas Ssp2 might regulate nuclear localization of Sre1N. The authors can address this comment by editing the text without additional experiments. The fact that this immunoblot does not perfectly repeat the microscopy experiment is actually quite interesting and reveals the potential for some specific roles for Ssp2 versus Gsk3/31 in regulating Sre1N. I encourage the authors to revise their interpretation to better reflect this nice result.

Response:

 Accordingly, we added these points in the revised manuscript (pages 3, lines 46-50 / page 5, lines 99-101 / page 15, line 310 / page 16, lines 334-344 / page 17, lines 353-354 / page 18, lines 379-384, 395 / page 25, lines 512-515).

We hope that these revisions would satisfy your requirements. Thank you very much for your time and consideration.

Sincerely,

Yue Fang, M.D, Ph.D.

Department of Microbial and Biochemical Pharmacy,

School of Pharmacy, China Medical University,

No.77 Puhe Road, Shenyang North New Area, Shenyang 110112, China

TEL: +86-18900910820, FAX: +86-24-31939448 

Email: yfang@cmu.edu.cn

---

## [Editor Report · Decision Letter 2]

27 Jan 2020

AMPKα Subunit Ssp2 and Glycogen Synthase Kinases Gsk3/Gsk31 are involved in regulation of sterol regulatory element-binding protein (SREBP) activity in fission yeast

PONE-D-19-30640R2

Dear Dr. Fang,

We are pleased to inform you that your manuscript has been judged scientifically suitable for publication and will be formally accepted for publication once it complies with all outstanding technical requirements.

Congratulations.

With kind regards,

Takashi Toda PhD

Academic Editor

PLOS ONE
---

## [Editor Report · Acceptance letter]

30 Jan 2020

PONE-D-19-30640R2 

AMPKα Subunit Ssp2 and Glycogen Synthase Kinases Gsk3/Gsk31 are involved in regulation of sterol regulatory element-binding protein (SREBP) activity in fission yeast 

Dear Dr. Fang:

I am pleased to inform you that your manuscript has been deemed suitable for publication in PLOS ONE. Congratulations! Your manuscript is now with our production department. 

With kind regards,

on behalf of

Prof. Takashi Toda PhD 

Academic Editor

PLOS ONE